# Peer review of "Primary Step Towards In Situ Detection of Chemical Biomarkers in the UNIVERSE via Liquid-Based Analytical System: Development of an Automated Online Trapping/Liquid Chromatography System"

_molecules, 2019, doi:10.3390/molecules24071429_

Round 1

Reviewer 1 Report

 The reviewed paper at title:

Primary step towards in situ detection of chemical biomarkers in the Universe via liquid-based analytical system: development of an automated online trapping/liquid chromatography system

This work presents the original configuration of liquid chromatography with a trapping unit designed to detect one set of large groups of non-volatile extraterrestrial compounds.

The paper seems to be acceptable (especially in this journal section) but, in my opinion, it requires some modifications. Additionally, several questions should be answered by the authors in detail, as many important issues are described too superficially:

Line 50-52: Are these not too bold proposals,      whether these are fully feasible, please specify the data for practical      information, possibly for already existing synonymous systems.

Line 76-79: I believe that it is worth expanding      these issues more accurately in the presented text.

Line 109: Please, improve the legibility of the      graphics.

Line 112-117: I am asking for more detailed      information on the retention time of oligopeptides.

Line 122: Please, improve the legibility of the      graphics.

Line 134: Is trimethionine the best choice ? Is      not it too sensitive, for example in the case of interaction with      proteases ?

Line 138: Please, improve the legibility of the      graphics.

Line 168-169: Please specify in terms of system      sensitivity.

Line 174: Please, improve the legibility of the      graphics.

Line 206: I think 3.2. Preparation of Standard      Solutions should be described a bit better, please clarify and broaden the      text.

In conclusion, the paper seems to be acceptable but requires some revisions. The whole layout and neatness of the paper do not leave too much objections, as it is prepared very carefully, but the quality of the discussion requires several amendments. Please answer all my questions and comments and attach the manuscript with marked changes. The objections presented by me do not undermine the quality of the paper, which will support in the further publishing process, certainly after careful consideration of my comments.

Author Response

We thank the reviewer for her/his comment.

Line 50-52: Are these not too bold proposals, whether these are fully feasible, please specify the data for practical information, possibly

Precisions were added.

As solid, liquid and aerosol samples are anticipated, the future instrument platform should involve versatile sample analysis. Gas and/or vapor samples can be analyzed directly by mass spectrometers or gas chromatograph-mass spectrometers [19,20] already successfully used in planetary [11,21] or cometary previous missions [10,22,23].

Line 76-79: I believe that it is worth expanding these issues more accurately in the presented text.

Precisions were added.

MS detectors coupled to liquid chromatography have already allowed main advances in organics characterization of meteorite, tholins and comets analogs [38–40]. Thanks to a Q-Orbitrap High Resolution Mass Spectrometer (HRMS) that allows a comprehensive assessment of the data obtained, several analytical features were assessed to define the best trapping and separation conditions. Finally, the approach combined with a simplified sample preparation protocol and detector about to be spatialized could potentially be validated for space life-search experiments.

Line 109: Please, improve the legibility of the graphics.

The graphics were improved to show the separation or coelution of the different targets.

Line 112-117: I am asking for more detailed information on the retention time of oligopeptides.

Retention times of oligopeptites eluted on Hypersil Gold AQ were added in Table 1.

Line 122: Please, improve the legibility of the graphics.

Scheme was simplified

Line 134: Is trimethionine the best choice? Is not it too sensitive, for example in the case of interaction with proteases?

Trimethionine was chosen due to its good ionization in MS and retention on the designed trapping system and the good retention on Hypersil Gold AQ. Trimethionine was an internal standard for trapping optimization but not defined for future space missions. As most of the peptides it will be sensitive to protease which is not however expected to be present in the samples.

Line 138: Please, improve the legibility of the graphics.

The graphics were improved to emphasize the separation and broadenings of the peaks of the different targets regarding trapping/ elution.

Line 168-169: Please specify in terms of system sensitivity.

Additional data showing the response of the designed system compared to a conventional laboratory system were adding in Figure 7

Line 174: Please, improve the legibility of the graphics.

The graphics were improved

Line 206: I think 3.2. Preparation of Standard Solutions should be described a bit better, please clarify and broaden the text.

Precisions were added.

The new statement in the revised manuscript is as follows:

Two stock solutions were prepared with oligopeptides solubilized in high purity water, amino acids in HCl 1 mol.L-1 and of nucleobases in NaOH 0.1 mol.L-1. The first one contained the peptides in high purity water (0.5 µg/mL), the second was composed of the amino acids and nucleobases (1.5 10-6 M). The standard solution containing the oligopeptides was used to prepare a working solution at 0.01 µg.mL-1, used to select the optimal trapping parameters of the trapping LC setup in comparison with the 1D-LC method. This one was further mixed and diluted with the standard solutions containing amino acids and bases to 6 calibration solutions injected in triplicates to validate the optimized system.

Reviewer 2 Report

The work is interesting and novel. I recommend its publication after some minor revisions.

The abstract is too simple and general, it sould be rewrited to highlight the background, the main research content, and the main results.

Secton 2 and 3 should be replaced each other.

About method validation, calibration curves, linearity range, recoveries and some other parameters are recommended providing.

Author Response

We thank Reviewer 2 for her/his kind comment.

The abstract is too simple and general, it should be rewrited to highlight the background, the main research content, and the main results.

Abstract: Search for biomarkers in our Solar System is a fundamental challenge for space research community. It encompasses major difficulties linked to their very low concentration levels, their ambiguous origins (biotic or abiotic) as well as their diversity and complexity. Even if in forty years, great improvements in sample pre-treatment, chromatographic separation and mass spectrometry detection have been achieved, there is still a need for new in situ scientific instrumentation. This work presents an original liquid chromatographic system with a trapping unit dedicated to the one pot detection of a large set of non-volatile extra-terrestrial compounds. It is composed of two units monitored by a single pump. The first unit is an online trapping unit able to trap polar, apolar, monomeric and polymeric organics. The second one is an analytical unit online with a high resolution Q-Orbitrap mass spectrometer. The designed single pump system was as efficient as a laboratory dual trap LC system for the analysis of amino-acids, nucleobases and oligopeptides. The overall setup significantly gains in sensitivity providing limits of detection ranging from ppb to ppt levels meeting thus with in situ enquiries.

Considering this recommendation, authors have decided to improve the abstract

Section 2 and 3 should be replaced each other.

Section 2 and 3 were reversed.

About method validation, calibration curves, linearity range, recoveries and some other parameters are recommended providing.

Additional data were added in table 2.

To further exemplify the sensitivity of the system when coupled to a mass spectrometer, the recovery of some targeted compounds was calculated using calibration curves (Table 2). Linearity ranged from 0.25 to 10 ng.mL-1.

Molecules EISSN 1420-3049 Published by MDPI AG, Basel, Switzerland RSS E-Mail Table of Contents Alert
Back to Top